# Enhancement of encoding and retrieval functions through theta phase-specific manipulation of hippocampus

**Joshua H Siegle[1,2]\*, Matthew A Wilson[1,2]\***

[1]Picower Institute for Learning and Memory, Massachusetts Institute of Technology, Cambridge, United States; [2]Department of Brain and Cognitive Sciences, Massachusetts Institute of Technology, Cambridge, United States

**Abstract** Assessing the behavioral relevance of the hippocampal theta rhythm has proven difficult, due to a shortage of experiments that selectively manipulate phase-specific information processing. Using closed-loop stimulation, we triggered inhibition of dorsal CA1 at specific phases of the endogenous theta rhythm in freely behaving mice. This intervention enhanced performance on a spatial navigation task that requires the encoding and retrieval of information related to reward location on every trial. In agreement with prior models of hippocampal function, the behavioral effects depended on both the phase of theta and the task segment at which we stimulated. Stimulation in the encoding segment enhanced performance when inhibition was triggered by the peak of theta. Conversely, stimulation in the retrieval segment enhanced performance when inhibition was triggered by the trough of theta. These results suggest that processes related to the encoding and retrieval of task-relevant information are preferentially active at distinct phases of theta.

**\*For correspondence:** jsiegle@mit.edu (JHS); mwilson@mit.edu (MAW)

**Competing interests:** The authors declare that no competing interests exist.

**Reviewing editor**: Howard Eichenbaum, Boston University, United States

## Introduction

Theta oscillations (4–12 Hz) are one of the most prominent rhythms in the mammalian brain (*Vanderwolf, 1969*; *Buzsáki et al., 1983*; *Colgin, 2013*). Theta is a distributed oscillation, which is broadly coherent between the left and right hippocampi, the entorhinal cortex, the medial septum, and various other cortical and subcortical recipients of hippocampal projections (*Buzsáki, 2002*). Neural activity is highly structured within each cycle of theta, with the firing rates of genetically defined cell types peaking at different phases (*Klausberger et al., 2003*, *2004*, *2005*). Spatially selective principal cells in the hippocampus fire at progressively earlier phases of theta as animals traverse their individual place fields, a phenomenon known as phase precession (*O'Keefe and Recce, 1993*; *Schmidt and Lipson, 2009*). Thus, the information content of hippocampal outputs changes throughout each cycle (*Mehta et al., 2000*, *2002*).

The organization of activity relative to theta appears to be important for behavior, since the degree to which other regions synchronize to the hippocampal theta rhythm is correlated with spatial decision-making performance (*Jones and Wilson, 2005*; *Sigurdsson et al., 2010*). But the specific role of theta in guiding behavior remains unclear, due to a lack of studies employing causal interventions with adequate temporal precision to selectively disrupt or enhance activity within this rhythm. Here, we employed a closed-loop approach to target an optogenetic manipulation to particular phases of endogenously generated theta oscillations. Closed-loop control is an under-utilized strategy for interrogating neural circuits, as it facilitates the testing of hypotheses that would be difficult or impossible to address through correlative methods (*Fetz, 1969*; *Rolston et al., 2010*; *Newman et al., 2012*; *Ngo et al., 2013*; *Paz et al., 2013*; *Wallach, 2013*).

**eLife digest** Around 15 years ago, an imaging study compared the brains of London taxi drivers—who need to know their way around one of the biggest cities in the world—with those of the general public, and found that a structure called the hippocampus was routinely larger in the taxi drivers. This finding was consistent with previous studies from rats, which showed that anatomical changes occur in the hippocampus after animals learn to navigate through various mazes. Together, these results suggest that the hippocampus is important for spatial awareness in both humans and rodents.

The hippocampus—which takes its name from the Greek for 'seahorse' due to its shape—consists mostly of cells called pyramidal neurons, which communicate with one other using an excitatory molecule called glutamate. However, it also contains cells that suppress the activity of the pyramidal neurons, using an inhibitory molecule called GABA. When electrodes are used to record the combined electrical activity of many cells in the hippocampus—including both excitatory and inhibitory cells—the resulting pattern resembles a wave with peaks and troughs that repeat roughly eight times per second. Although this activity, known as the theta rhythm or cycle, has been observed in countless experiments, it has been difficult to pin down how it is relevant to behavior.

Siegle and Wilson now show that the theta cycle may help the brain to keep incoming information separate from information stored in memory. This conclusion is based on the results of experiments on mice with hippocampi that had been modified to make them sensitive to light: in particular, light was needed to activate the neurons that suppress the activity of the pyramidal neurons. This meant that it was possible to reduce the overall level of activity in the hippocampus by shining light on certain neurons.

The mice were trained to perform a spatial memory task that consisted of an encoding stage—where they learned the location of a reward—and a retrieval stage, in which they recalled this location from memory. On certain trials, pulses of light could be delivered to the brain at specific points in the theta cycle. Delivering light near the peak of the cycle during the encoding stage resulted in improved memory performance, as did delivering light near the trough of the cycle during the retrieval stage.

These results suggest that the hippocampus preferentially encodes and retrieves information at different stages of the theta cycle. Specifically, activity just after the peak of the theta cycle is biased towards retrieval, meaning that reducing hippocampal activity at this time point will make it easier to form new memories. By contrast, reducing activity just after the trough of the theta cycle—when the hippocampus is biased towards encoding—will enhance memory retrieval.

One specific hypothesis about the mnemonic role of theta is that it partitions processes related to the encoding of new information and the retrieval of stored information (*Hasselmo et al., 2002*). In order to carry out their roles in spatial navigation, the hippocampus and related structures must be able to distinguish activity that tracks the current state of the world from activity that reflects prior experience. Theta could serve to coordinate cell assemblies such that encoded and retrieved information are less likely to interfere (*Hasselmo et al., 2002*; *Hasselmo, 2005*; *Colgin and Moser, 2010*).

There is abundant correlative evidence that inputs to the hippocampus vary as a function of theta phase. Input from the entorhinal cortex (EC), the major source of cortical projections to the hippocampus, is highest at the trough of theta waves recorded at the hippocampal fissure (*Brankack et al., 1993*; *Kamondi et al., 1998*). Because it conveys information about the outside world, this input is likely associated with encoding of the current state of the environment (*Hasselmo, 2005*). At this same phase, the hippocampus is more susceptible to long-term potentiation (*Hyman et al., 2003*; *Kwag and Paulsen, 2009*), consistent with the idea that this phase is optimized for encoding new information. At the 180° phase offset (the peak of fissure theta), CA1 cells receive greater input from upstream cells in CA3 (*Hasselmo, 2005*). At this phase, stimulation of Schaffer collateral or temporoammonic inputs induces long-term depression (*Hyman et al., 2003*; *Kwag and Paulsen, 2009*), which could suppress information storage during memory retrieval. As a result of these phase-specific physiological changes, hippocampal networks can regulate the behavioral impact of different types of information as a function of task context.

Simultaneously recording in CA1, CA3, and EC reveals that oscillations in the high gamma range (60–100 Hz) are a signature of enhanced coordination between CA1 and EC, whereas low gamma (25–50 Hz) indicates enhanced coordination between CA3 and EC. Furthermore, these oscillations occur at different phases of theta, and typically on different cycles (*Colgin et al., 2009*). Taken together, these results indicate that the balance between the relative influence of the outside world (via EC) and internal states (via CA3) on hippocampal outputs is strongly modulated as a function of theta phase. Other studies have observed task-dependent modulation of hippocampal firing that is consistent with encoding of novel stimuli and retrieval of stored memories being biased to different phases (*Manns et al., 2007*; *Lever et al., 2010*; *Douchamps et al., 2013*; *Newman et al., 2013*). Computational modeling studies provide further support for this hypothesis (*Hasselmo et al., 2002*; *Hasselmo and Eichenbaum, 2005*; *Kunec et al., 2005*).

These results do not imply that new information is encoded and stored information is retrieved on every cycle of theta (~8 Hz). When encoding and retrieval do occur, though, they may be preferentially active at different phases, to take advantage of the temporal structure of activity within the hippocampal–EC loop (*Mizuseki et al., 2009*; *Colgin and Moser, 2010*). Thus, a manipulation that targets a specific phase of theta could, on average, selectively modulate one process or the other.

If encoding and retrieval processes are most active at different times within the theta cycle, the consequences of a phase-specific intervention should depend on the behavioral context. Manipulations that alter hippocampal outputs at one phase of theta may have a strong impact on behavior if they occur while information is being encoded, but no effect (or the opposite effect) if they occur while information is being retrieved. Conversely, manipulations that occur with a 180° phase offset may have their behavioral impact limited to intervals in which retrieved information is used to guide behavior, but have no effect (or the opposite effect) at times when task-relevant information is being encoded.

In this study, we used millisecond-timescale optogenetic control of intrinsic inhibition to gate hippocampal outputs at specific phases of the ongoing theta cycle. Although previous studies have used optogenetic interventions to highlight the role of inhibition in phase precession (*Royer et al., 2012*) and theta resonance (*Stark et al., 2013*), here the goal was to suppress firing of CA1 in a phase-specific manner. We directly activated parvalbumin-positive interneurons, which deliver fast and powerful inhibition to the cell bodies of pyramidal neurons in the hippocampus (*Bartos et al., 2007*), at either the falling phase or rising phase of theta recorded in the local field potential (LFP). We performed LFP-phase-triggered optogenetic feedback in the context of a spatial navigation task, in which mice must encode and retrieve location information on every trial (*Jones and Wilson, 2005*). Our stimulation occurred relative to the phase of locally recorded theta on the trigger electrodes, rather than the phase at the hippocampal fissure, to which much of the previous literature uses as a landmark (*Brankack et al., 1993*; *Kamondi et al., 1998*; *Hasselmo et al., 2002*). However, post-hoc analysis revealed that light pulses were delivered at similar absolute phases across animals.

Using closed-loop optogenetics to intervene on the timescale of theta oscillations is a powerful approach. It allows us to alter hippocampal outputs relative to ongoing theta rhythms on a trial-by-trial basis, providing within-animal controls for all stimulation conditions. We found that triggering inhibition on the peak of theta improved performance when it occurred in the encoding segment of the task, but had no effect in the retrieval segment. Triggering inhibition on the trough of theta had the opposite effects: it enhanced retrieval performance, but did not affect encoding processes.

## Results

### Mice learn to perform a spatial navigation task

We trained mice on a navigation task that required encoding and retrieval of reward location on individual trials. Mice were placed on an H-shaped track, which consisted of two choice points separated by a central arm (*Jones and Wilson, 2005*) (*Figure 1A*). At one junction, a movable barrier forced mice to make a left or right turn in order to arrive at the start location. At the other junction, mice were free to turn in either direction. A food reward was delivered only if mice chose the arm closest to the most recent start location.

In order to perform the task above chance, mice must update their knowledge of reward location on a trial-by-trial basis. During the encoding segment of the task (start arms), environmental cues signal the location of the upcoming reward. During the retrieval segment of the task (central arm), information about the start arm is no longer present, and thus activity that drives decision-making

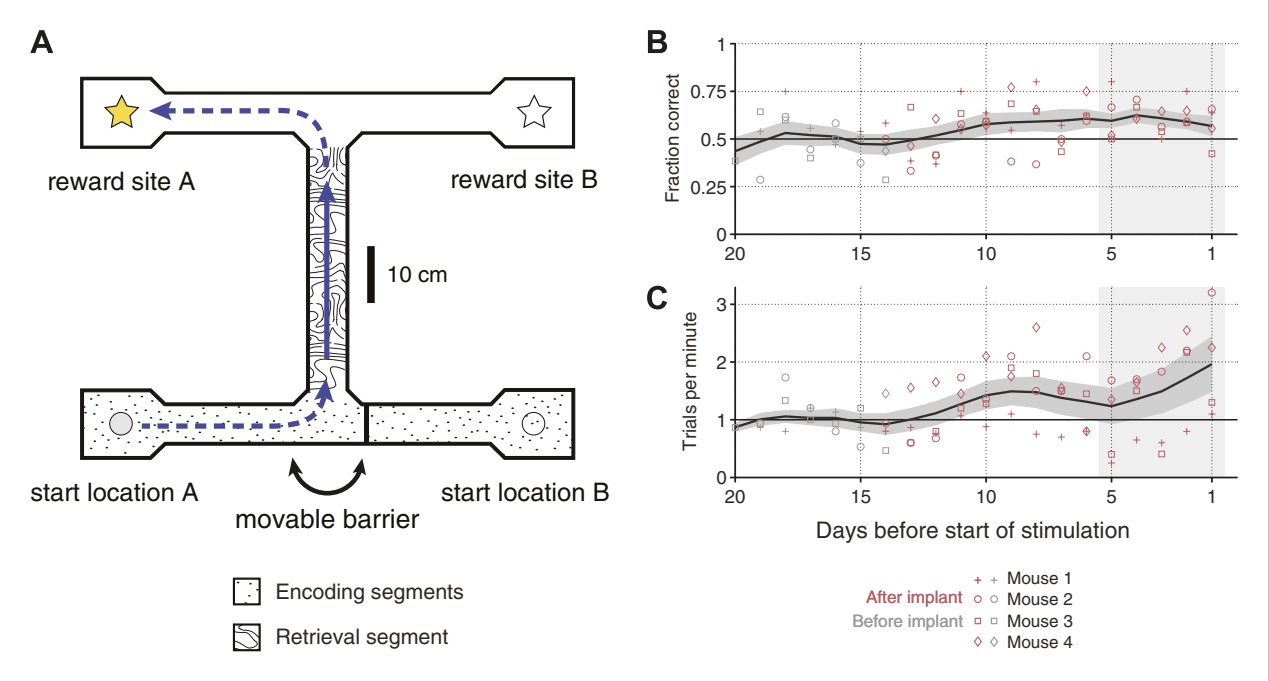

**Figure 1**. Overview of the behavioral task. (**A**) Scale drawing of the end-to-end T-maze used in all experiments. On each trial, mice navigate through the 'retrieval segment' in the direction of the solid arrow and must choose between one of two reward sites. Reward is delivered for trajectories that involve two turns in the same direction (e.g., the 'left/left' trajectory shown). Once the reward site is reached, mice must travel back to one of two start locations in order to initiate the next trial. A movable barrier determines the start location for that trial, and hence which reward site will contain the food pellet. The barrier is repositioned randomly after each visit to a reward site (whether correct or incorrect). A second barrier (not shown) prevents mice from navigating between reward sites after a decision has been made. The maze is surrounded by 10 cm walls made of clear acrylic, through which distal cues are visible. (**B**) Fraction of correct trials in each session leading up to the start of optogenetic stimulation for *N* = 4 individual mice (open shapes) and the mean ± SEM. across all subjects (5-day running average). In the 5 days before the start of optogenetic stimulation (shaded region), all mice perform significantly above chance (p<0.05, based on p.d.f. of the binomial distribution with probability of 0.5). (**C**) Trials per minute for the same sessions as in **C**. Mean for last 5 days (shaded region) is 36.1 ± 18.3 trials per session per mouse.

must be generated internally. This task makes it possible to dissociate the effects of theta phase–specific inhibition on encoding and retrieval processes by separating the cues to reward location from the time and location of the mouse's decision.

Four mice were trained on this task over the course of 2 to 4 weeks. All mice expressed the gene for Cre-recombinase in parvalbumin-positive cells, to allow us to target expression of channelrhodopsin to these cells later in the experiment. In the last 5 days of training, all mice performed at levels significantly above chance (p<0.05, based on p.d.f. of binomial distribution with chance level of 0.5), with an average probability of correct response of 0.61 ± 0.05 (*Figure 1B*). In addition to improving their accuracy, mice also increased the speed at which they performed the task, to 1.49 ± 0.69 trials per minute during the last 5 days of training (*Figure 1C*).

## Recruiting fast inhibition as a function of ongoing theta phase

After at least 8 days of training, mice were implanted with a multielectrode array that targeted movable tetrodes and stationary fiber optic cables to hippocampus bilaterally. Two fiber optic cables (one per hemisphere) were implanted to a depth of 0.9 mm at the time of surgery. In the same procedure, we injected 1.0 μl of an adeno-associated virus carrying the gene for channelrhodopsin-2 (*Nagel et al., 2003*) into both sides of the brain, centered on CA1 approximately 1 mm posterior to the septal pole of hippocampus. Expression spread at least 2 mm along the septotemporal axis, covering most of dorsal CA1 as well as overlying cortex (*Figure 2A*).

We waited at least 2 weeks for ChR2 expression levels to increase, during which we lowered electrodes toward the hippocampus and continued to train animals to criterion. During test sessions, we used a 465 nm LED light to drive parvalbumin-positive interneurons, which are primarily fast-spiking,

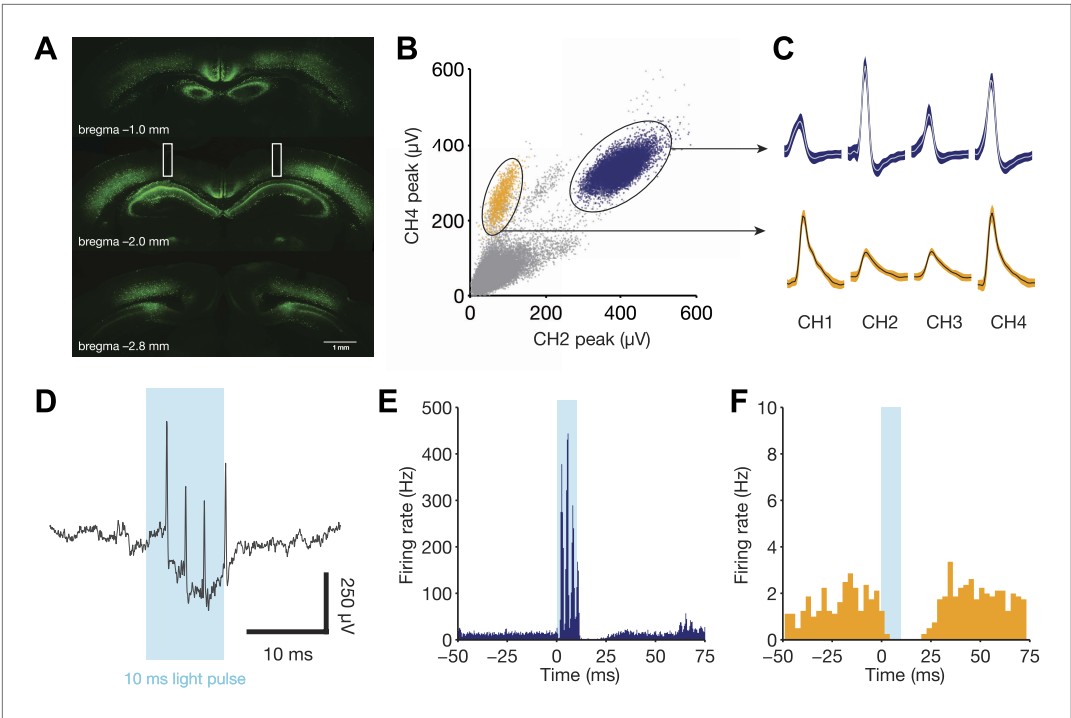

**Figure 2**. Direct recruitment of fast-spiking inhibition with light. (**A**) Expression of ChR2-EYFP throughout the dorsal hippocampus. Note the strong labeling in stratum pyramidale, indicative of dense PV+ projections in this layer. Bilateral fiber optic lesions are marked with white rectangles, centered at ~2 mm posterior to bregma and ~1.75 mm lateral to the midline. (**B**) Projection plot of peak heights from a CA1 electrode containing a well-isolated fast-spiking unit (blue) and a well-isolated regular-spiking unit (yellow). (**C**) Mean waveforms (with SD) for each tetrode channel for the same units as in panel **B**. (**D**) Raw, broadband trace for a single trial, aligned to the 10 ms light pulse. Four light-evoked spikes from the fast-spiking unit are clearly identifiable. (**E**) Peri-stimulus time histogram for the fast-spiking unit in **B**, **C**, and **D**, aligned to the start of each light pulse ($N = 1106$ pulses from one session). This unit responds with 3–4 spikes per stimulus, then remains silent for a period of ~15 ms following light offset. (**F**) Peri-stimulus time histogram for the regular-spiking unit in **B** and **C**, aligned to the start of each light pulse ($N = 1106$ pulses from one session). This unit is silenced for ~25 ms following light onset.

soma-targeting basket and chandelier cells in the hippocampus (*Pawelzik et al., 2002*). All light pulses lasted 10 ms and had an irradiance of 50 mW/mm² (~2.5 mW from a 250 micron fiber optic cable). Individual pulses reliably elicited up to four spikes from well-isolated fast-spiking units (peak rate of 400 Hz, *Figure 2B–E*). Nearby regular-spiking units were inhibited for a period of 25 ms following light onset (*Figure 2F*), consistent with the known time constant of fast-spiking inhibition (*Bartos et al., 2007*).

We combined optogenetic stimulation with closed-loop feedback in order to trigger inhibition at specific phases of theta. In each mouse ($N = 4$), an electrode with high theta power in the local field potential was chosen as the 'trigger' channel. The signals from these electrodes were filtered between 4 and 12 Hz in software. When the signal reached a local maximum or minimum, the software triggered a 10 ms light pulse delivered simultaneously to both implanted fiber optic cables (*Figure 3A,B*). A light pulse was delivered once per theta cycle as long as the mouse remained in the stimulation zone. The same trigger channel was used throughout the course of the experiment.

Within an individual session, stimulation was confined to the retrieval (middle arm) or encoding (sample arms) segments of the track (*Figure 1A*). In the *retrieval* segment, mice run toward the choice point. Stimulation in this region may affect the retrieval of stored information about reward location, but not encoding of information directly relevant for task performance. In the *encoding* segments, mice explore one of two sample arms. In these regions, stimulation could affect the encoding of available information about reward location. In both cases, however, the behavioral readout is the same: whether or not the mouse turned in the correct direction to retrieve the reward for that trial.

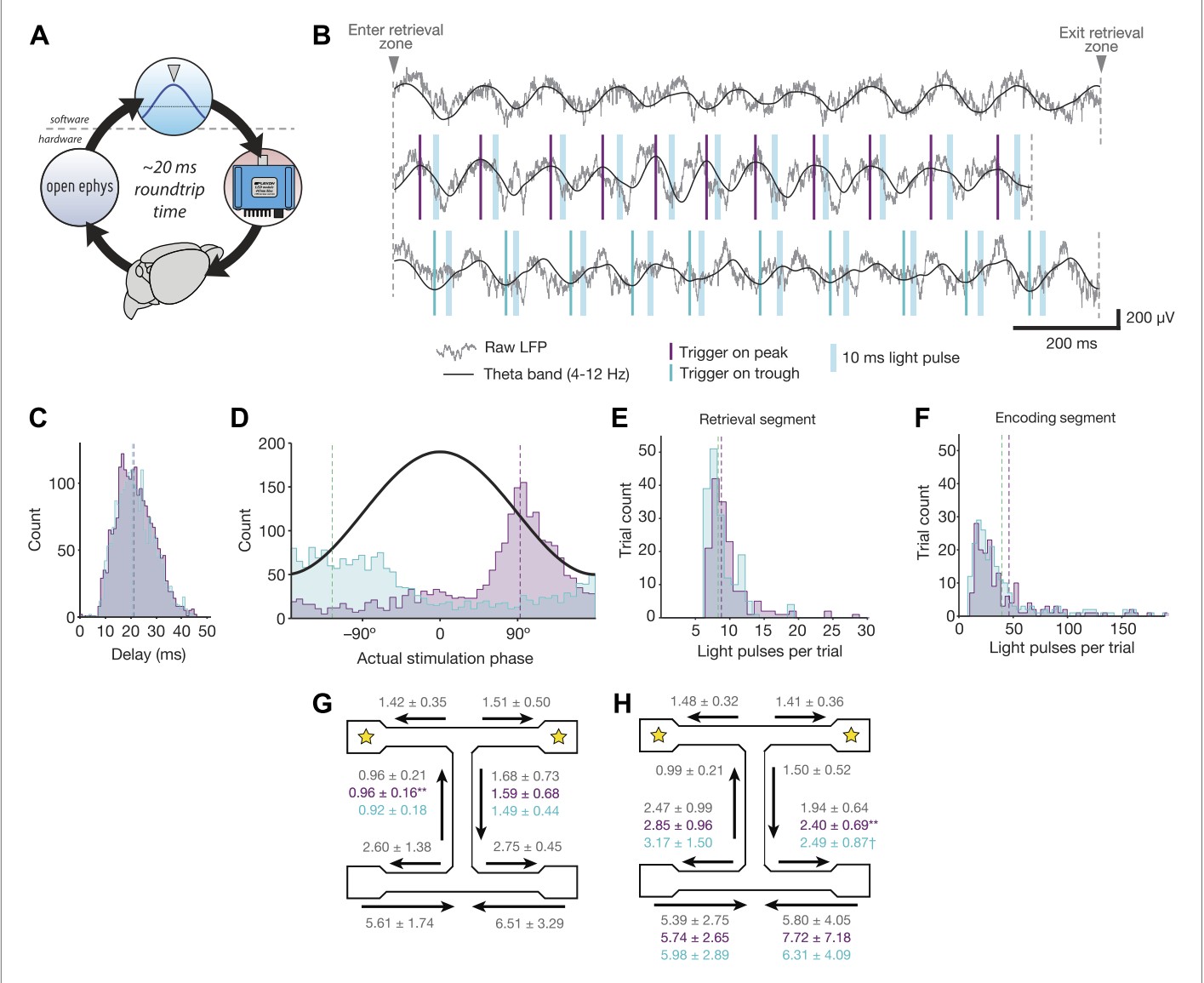

**Figure 3**. Properties of theta-triggered stimulation. (**A**) Schematic of steps involved in delivering closed-loop feedback. An event occurs in the brain (bottom), which is detected and digitized by the Open Ephys recording hardware (left), and sent to software for analysis (top). When the target event is detected, the software activates an LED (right) which delivers light to brain via implanted fiber optic cables (bottom). (**B**) Examples of raw and theta-bandpassed LFP during baseline trials (top), peak-triggered stimulation trials (middle), and trough-triggered stimulation trials (bottom). Vertical blue bars indicate the time at which 10 ms light pulses occur on each cycle. (**C**) Distribution of delays between detection of the actual theta peak (purple) or trough (teal) and the time of stimulus delivery. (**D**) Distribution of actual theta phases at which stimulation occurred, for both peak (purple) and trough (teal) trials. Peak-triggered stimulation tends to occur during the falling phase of theta, whereas trough-triggered stimulation occurs around the actual trough and rising phase. Phase was calculated for data filtered offline between 4 and 12 Hz, to eliminate the phase delays inherent in online filtering. (**E**) Distribution of pulses per trial for the retrieval segment of the track. (**F**) Same as **E**, but for encoding segments of the track. (**G**) Occupancy times in different segments of the track for trials with retrieval-segment stimulation. Values for peak-triggered and trough-triggered stimulation are shown in purple and teal, respectively (mean ± SD for *N* = 4 mice; ** = occupancy time decreased significantly for one mouse, p<0.005, Wilcoxon rank sum test with Bonferroni correction). (**H**) Occupancy times in different segments of the track for trials with encoding-segment stimulation. Values for peak and trough-triggered stimulation are shown in purple and teal, respectively (mean ± SD for *N* = 4 mice; ** = occupancy time decreased significantly for one mouse, p<0.005; † = occupancy time increased significantly for one mouse, and decreased significantly for a different mouse, p<0.05; Wilcoxon rank sum test with Bonferroni correction).

Individual trials were classified as one of three types: baseline (no stimulation), peak-triggered stimulation, or trough-triggered stimulation (*Figure 3B*). All trials types were randomly interleaved and occurred with equal probability. Three mice experienced the retrieval stimulation condition first,

followed by the encoding stimulation condition. One mouse experienced the conditions in the opposite order. Analysis was limited to the first 150 trials for each condition (encoding or retrieval).

The properties of our closed-loop stimulation were as follows: the mean delay between the trigger event (peak or trough of theta reached) and the onset of the light pulse was 21.7 ± 7.2 ms for peak-triggered stimulation and 21.3 ± 7.4 ms for trough-triggered stimulation (*Figure 3C*). This is equivalent to approximately 1/6 of a 125 ms theta cycle. The mean phase of stimulation (based on offline-filtered theta with no phase delay) was 96 ± 54° for peak-triggered stimulation and −131 ± 63° for trough-triggered stimulation (*Figures 3E* and 0° = peak). The phase targeting for trough-triggered stimulation was less precise, as the rising phase of theta is shorter than the falling phase (*Belluscio et al., 2012*). The mean number of pulses per trial in the retrieval-stimulation condition (middle arm) was 8.8 ± 3.3 for peak-triggered stimulation and 8.3 ± 8.3 for trough-triggered stimulation. For the encoding-stimulation condition, the mean number of pulses was 46.1 ± 57.2 for peak-triggered stimulation and 39.6 ± 39.6 for trough-triggered stimulation (*Figure 3D–F*).

Stimulation did not generally alter occupancy time in different segments of the track (*Figure 3G–H*). On each trial, mice spent the majority of time in the encoding segment (average of 2–3 s for inbound trajectories and 5–8 s for outbound trajectories). Once they left the sample arm, they ran quickly toward the goal, spending 1–2 s in the retrieval segment and a similar amount of time running toward the reward location after making their decision. The addition of optogenetic feedback only changed occupancy times significantly for one mouse in the retrieval and outbound encoding segments, and a second mouse in the outbound encoding segment. Otherwise, all occupancy times were similar (p>0.05, Wilcoxon rank sum test with Bonferroni correction for two tests per segment, $N \geq 44$ trials per segment per mouse).

Optogenetic feedback altered the average power spectrum across trigger channels, for example by increasing the peak frequency and amplitude of theta during the peak-triggered stimulation condition (*Figure 4A*). There was also an increase in power in the low-gamma band (25–35 Hz) for both peak and trough stimulation, but this was associated with a much stronger peak in the beta band (16–25 Hz), which may have affected the low-gamma band via spectral leakage. Based on the shape of the evoked response to each optogenetic stimulus, it appears that these effects are due to the frequency content of the average waveform, rather than non-phase-aligned induced power in different frequency bands (*Figure 4B*). Aligning the local field potential to the start of each light pulse revealed a large deflection, 200–400 µV in amplitude. The shape of the average response accounts for both the shifts in theta frequency (based on the location of the subsequent peak), and the beta-range power increases (due to ~50 ms deflections). Individual pulses affected the amplitude of subsequent cycles of theta, as evidenced by the difference in the mean LFP between −100 and −75 ms for actual (purple and teal) vs dummy (gray) stimulation conditions.

Optogenetic stimulation was always aligned to the relative peak or trough of the 4–12 Hz band-passed signals on each trigger electrode (*Figure 3D*). To permit meaningful interpretation of the analysis of our behavioral results, it was necessary to measure the time of stimulation relative to an absolute indicator of theta phase. We chose high gamma (60–80 Hz) power, which showed strong phasic modulation across all hippocampal electrodes, and has been previously shown to occur at a consistent phase of theta (*Colgin et al., 2009*). Therefore, the peak of high gamma on baseline trials served as a landmark within each cycle of theta. In 3/4 mice, we measured stimulation times relative to the peak of high gamma for both the trigger electrode and a neighboring electrode that was passively recording signals (*Figure 4C*). Although post-mortem analysis of electrolytic lesions revealed different locations for each electrode, all electrodes indicated that peak-triggered stimulation occurred just after the trough of high gamma, whereas trough-triggered stimulation occurred around or after the high gamma peak. In one mouse, we could not measure absolute stimulation phase, due to the trigger electrode's location in L5/6 of cortex overlying hippocampus. This electrode expressed high theta power, presumably volume-conducted from hippocampus, which was used to trigger stimulation. However, it lacked associated high gamma power, which occurs more locally.

The consistency of this result indicates that, despite variations in electrode location, absolute stimulation phase was similar across animals. Although we did not measure CA1–MEC synchronization directly, previous studies have shown high gamma power to be a reliable indicator of enhanced coordination between these regions (*Colgin et al., 2009*; *Yamamoto et al., 2014*). Therefore, we hypothesize that trough-triggered stimulation resulted in optogenetic stimulation occurring during phases of theta in which CA1–MEC coordination was high, thereby providing CA1 with access to information

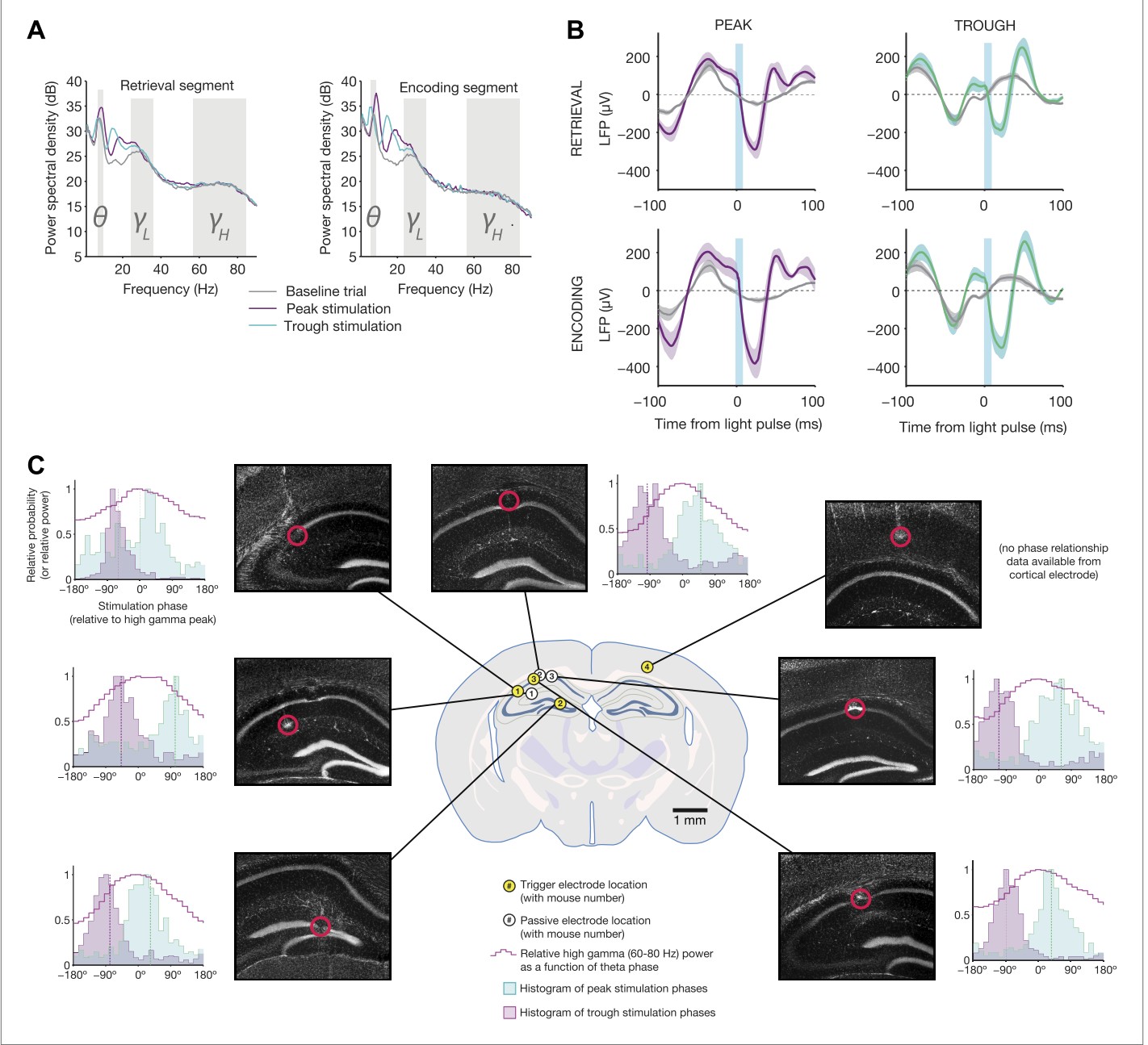

**Figure 4**. Electrophysiological changes induced by theta-triggered stimulation. (**A**) Mean power spectra for baseline, peak-triggered stimulation, and trough-triggered stimulation trials, while mice were in the retrieval segment heading toward the reward arm (left) or the encoding segment prior to entering the trial start location at the end of the sample arm (right) (*N* = 4 electrodes from four mice used for triggering online feedback). Theta, low gamma, and high gamma frequency bands are highlighted. (**B**) Average light-evoked LFP response from *N* = 3 hippocampal electrodes for peak and trough-triggered stimulation trials (purple and teal traces, respectively), for both encoding and retrieval epochs (mean ± SEM). Gray traces indicate the average theta waveforms for baseline trials, aligned to the time a stimulus would have occurred, but for which no actual light pulse was present. (**C**) Locations of trigger electrodes (yellow) and passive recording electrodes (white) for four mice used in this experiment. The location of each lesion is indicated by red circles superimposed over histological sections (DAPI stain, grayscale image of blue channel). Next to each of the images is a histogram of peak and trough stimulation phases, relative to the peak of high gamma power on that electrode for baseline (no stimulation) trials (indicated by 0°). High gamma power (a signature of synchronization between hippocampus and medial entorhinal cortex (*Colgin et al., 2009*), provides an absolute indication of theta phase, against which the time of our optogenetic stimulation can be compared. In all electrodes (except for the one trigger electrode in cortex, where high gamma was not measured), trough stimulation occurs after the peak of high gamma power, while peak stimulation occurs before the peak of high gamma.

about the current state of the world (*Hasselmo et al., 2002*; *Hasselmo and Eichenbaum, 2005*; *Colgin et al., 2009*; *Colgin and Moser, 2010*). Peak stimulation, on the other hand, targeted stimulation to phases in which CA1 and CA3 are most active (*Mizuseki et al., 2009*), during which information from the hippocampus can drive downstream structures.

## Impact of closed-loop inhibition on behavior depends on both theta phase and task segment

The effects of closed-loop optogenetic feedback on behavior depended on both the phase of theta used to trigger stimulation and the region of the track in which the stimulation occurred. On individual trials, 10 ms light pulses were triggered on either the peak or trough of theta (*Figure 5A*, phase relative to theta at the hippocampal fissure). When stimulation occurred in the retrieval segment, performance did not differ between baseline and peak-triggered stimulation for 4/4 mice (mean of 57.3 ± 10.0% correct for baseline vs 57.8 ± 10.5% correct for peak, individual results in *Table 1*). For trough-triggered stimulation, however, performance improved significantly in 4/4 mice (71.0 ± 8.2% correct for trough; significance determined by the p.d.f. of the binomial distribution, with baseline accuracy for each mouse used as the 'chance' level). The opposite effects were observed for stimulation in the encoding segment. In this condition, performance during trials with trough-triggered stimulation did not differ from baseline in 3/4 mice (mean of 59.1 ± 2.4% correct for baseline vs 58.7 ± 10.5% correct for trough; 1 mouse had significantly impaired performance in the trough-stimulation condition). For peak-triggered stimulation, performance improved significantly in 3/4 mice (mean of 69.6 ± 14.7% correct; 1 mouse showed no difference from baseline).

On average, trough-triggered stimulation resulted in a 13.7% improvement in accuracy for the retrieval condition, while peak-triggered stimulation resulted in a 10.5% improvement in accuracy for the encoding condition (*Figure 5B*). The effects were consistent across individual mice, with trough-triggered stimulation improving performance more for the retrieval segment than the encoding segment in 4/4 mice, and peak-triggered stimulation improving performance more for the encoding segment than the retrieval segment for 3/4 mice (*Figure 5C*). Such effects represent a double-dissociation, as phase-specific optogenetically recruited inhibition reversed its behavioral impact depending on the region of stimulation.

To estimate the probability that these results could have occurred by chance, we had to consider all outcomes in which a double dissociation was present. Our initial hypothesis was only that the effects of stimulation would depend on both task phase and theta phase, not that performance would be specifically impaired or improved. We used a bootstrap procedure with 10,000 repetitions to determine the probability that any of the possible double dissociations shown in *Figure 5D* could have occurred by chance. We randomized the labels for all trials (baseline, peak-triggered, and trough-triggered) and looked for the presence of significant changes relative to baseline in any of the conditions in the 2 × 2 square. If 3/4 mice showed the same behavior (enhancement, impairment, or no change), we considered that a 'consistent' quadrant. The probability that the same effects would be seen along any diagonal was 0.0013. The probability that the same effects were seen along *both* diagonals (what we observed in the actual data) was 0.0001.

To better understand the source of these behavioral effects, we analyzed the types of mistakes made by the mice, and how activating inhibition at the appropriate phase serves to correct them. It is clear that the outcome of the previous trial has a strong effect on decision-making: mice are much more likely to make a correct choice if they are cued to switch arms after failing to receive reward (82 ± 14% correct) or return to the same arm after receiving reward (72 ± 18% correct; mean response across encoding and retrieval conditions, baseline trials only). They are less likely to switch arms after a correct decision (32 ± 9% correct) or to return to the same arm after an incorrect choice (40 ± 21% correct). Thus, mice exhibit a bias toward a 'win-stay, lose–switch' strategy. They favor returning to arms in which they just received reward, or switching to the opposite arm if there was no reward on the previous trial (*Figure 5E*).

Adding optogenetic stimulation on certain trials allows mice to overcome this inherent bias. The strongest effects were seen for trials in which mice were cued to enter the opposite reward arm after making a correct choice. On average, trough-triggered stimulation in the retrieval segment improved performance on these trials by 19.5 ± 7.3%, with improvement seen in 4/4 mice (versus 5.1 ± 14.7% for peak-triggered stimulation). Peak-triggered stimulation in the encoding segment improved performance on these trials by 25.0 ± 10.3%, again with improvement in 4/4 mice (versus 8.7 ± 14.6% for

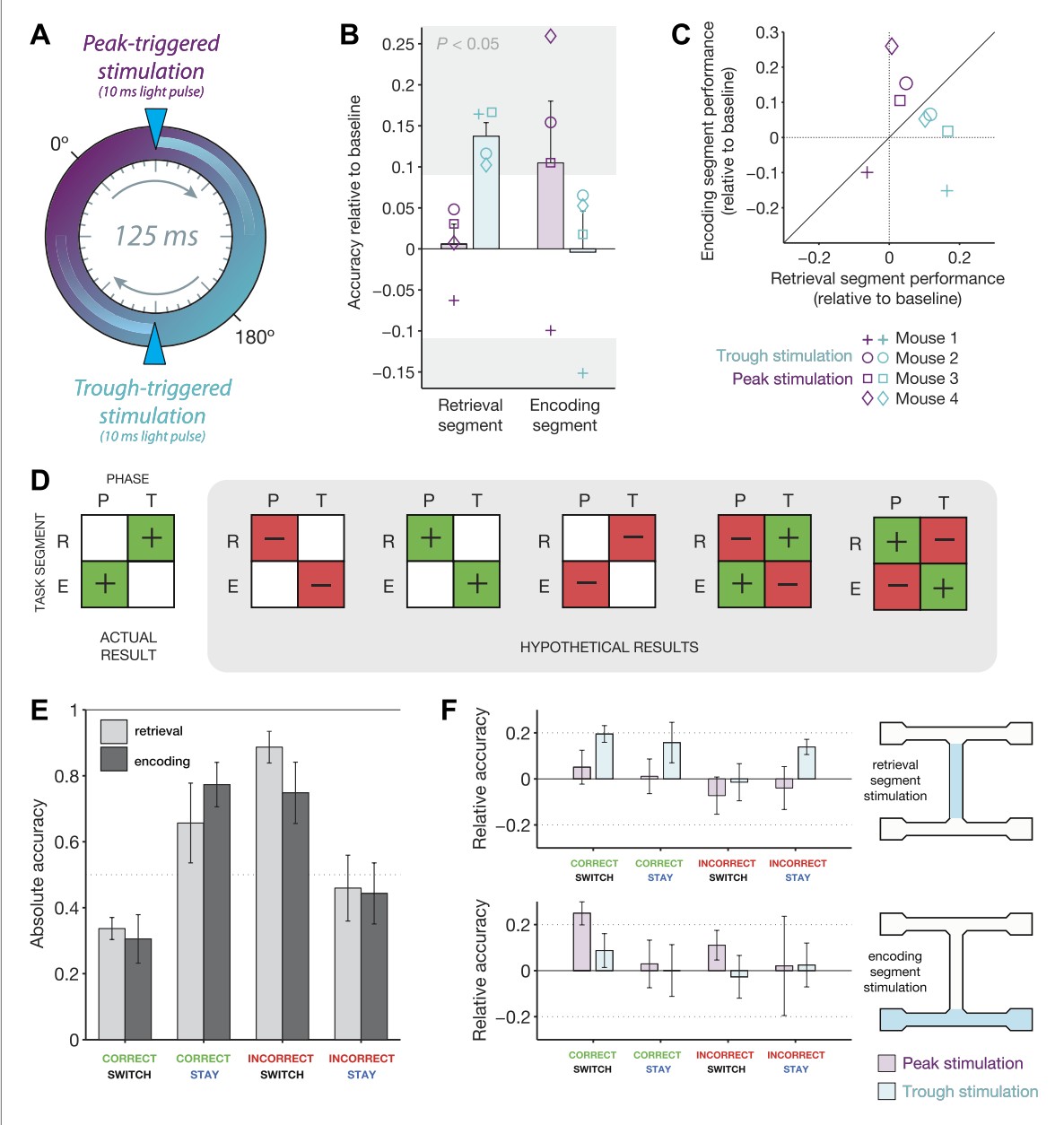

**Figure 5**. Behavioral modulation depends on both theta phase and task segment. (**A**) Illustration of the two manipulations performed in this experiment. On any given 'non-baseline' trial, stimulation was triggered by the peak (purple phase) or trough (teal phase) of the 4–12 Hz theta rhythm. The resulting light pulses recruited inhibition for ~25 ms, or approximately 1/5 of the 125 ms theta cycle. (**B**) Accuracy relative to baseline for four mice in four conditions: optogenetic stimulation triggered at the peak (purple) or trough (teal) of theta, in either the retrieval (left) or encoding (right) segments of the track. Mean ± SEM, with results for each mouse overlaid. Individual results in the gray regions are significantly different from baseline (p<0.05, p.d.f. of binomial distribution with probability equal to baseline accuracy). (**C**) Same data as in b, but represented on the same axes. Note that peak-triggered stimulation in the encoding segment consistently improves performance more than the same type of stimulation in the retrieval segment (points above diagonal line). The opposite effects are seen for trough-triggered stimulation. (**D**) Schematic of all possible 'double-dissociation' scenarios used for establishing bootstrap significance levels of the actual result. (**E**) Performance on baseline (no stimulation) trials for four different trial types: (1) mice are cued to switch arms after a correct choice (correct/switch), (2) mice are cued to return to the same arm after a correct choice (correct/stay), (3) mice are cued to switch arms after an incorrect choice (incorrect/switch), and (4) mice are cued to return to the same arm after an incorrect choice. Trials are grouped by retrieval stimulation or encoding stimulation conditions. For both conditions, changing trial type has a significant effect on performance: retrieval stimulation, $\chi^2$ = 8.4, p=0.038; encoding stimulation, $\chi^2$ = 8.1, p=0.044; Friedman test (nonparametric, repeated-measures ANOVA). (**F**) Change in performance with the addition of closed-loop optogenetic stimulation for the four trial types in **E**.

**Table 1.** Results for individual mice

| | Retrieval | | | Encoding | | |
| --- | --- | --- | --- | --- | --- | --- |
| | **Baseline** | **Peak** | **Trough** | **Baseline** | **Peak** | **Trough** |
| Mouse 1 | 0.55 | 0.49 | 0.71 | 0.58 | 0.48 | 0.43 |
| | | p=0.09 | **P=0.02** | | p=0.07 | **P=0.02** |
| Mouse 2 | 0.52 | 0.57 | 0.64 | 0.60 | 0.75 | 0.66 |
| | | p=0.09 | **P=0.02** | | **P=0.01** | p=0.06 |
| Mouse 3 | 0.50 | 0.53 | 0.67 | 0.62 | 0.73 | 0.64 |
| | | p=0.10 | **P=0.01** | | **P=0.03** | p=0.11 |
| Mouse 4 | 0.72 | 0.73 | 0.82 | 0.56 | 0.82 | 0.62 |
| | | p=0.12 | **P=0.04** | | **P=0.0001** | p=0.08 |

Probability of a correct response for four mice under six conditions: baseline (no stimulation), peak-triggered stimulation, and trough-triggered stimulation in both the retrieval and encoding segments. p-values computed from the p.d.f. of the binomial distribution with chance levels equal to the baseline performance for that mouse. Significant changes (p<0.05) are highlighted in bold.

trough-triggered stimulation). The effects of phase-specific stimulation on other types of errors were less pronounced, but there was no evidence for reduced performance by the 'optimal' stimulation phase for any trial type (*Figure 5F*).

## Discussion

These results provide new evidence for a hypothesis that was previously supported by correlational studies and computational models: processes related to encoding new information and retrieving stored information occur preferentially at different phases of the theta oscillation (*Hasselmo et al., 2002*; *Hasselmo, 2005*; *Colgin et al., 2009*; *Colgin and Moser, 2010*). We have shown that interventions targeting the falling or rising phases of theta have different effects depending on the behavioral context. When environmental cues to reward location are available (as in the encoding segment of the task), triggering hippocampal inhibition on the peak of theta enhanced navigational accuracy. When behavioral guidance must be based on internal signals alone (as in the retrieval segment of the task), triggering hippocampal inhibition on the trough of theta increased the probability of a correct choice.

What is the neural basis these effects? Our favored explanation is that phase-specific inhibition serves to reduce the response to task-irrelevant inputs. *Figure 6*, which was inspired by a similar diagram in *Hasselmo et al. (2002)*, illustrates the mechanism by which this could occur. On average, the influence of CA3 and entorhinal cortex inputs to CA1 changes as a function of theta phase (*Hasselmo et al., 2002*). Under baseline conditions, the relative influence of CA3 and EC is 'balanced'. With the addition of closed-loop optogenetic feedback, excess inhibition reduces spike activity either during EC-dominant or CA3-dominant periods of the theta cycle. Although the parvalbumin-positive interneurons recruited by this manipulation are typically active during the CA3-dominant phases of theta (*Lasztóczi and Klausberger, 2014*), causal control allows us to activate them synchronously at arbitrary times during the theta cycle. Under the proposed mechanism, inhibiting CA1 during EC-dominant cycles in the retrieval segment improves task performance by increasing the relative influence of CA3. Conversely, inhibiting CA1 during CA3-dominant cycles in the encoding segment improves task performance by increasing the relative influence of EC or by suppressing retrieval of interfering cross-trial information. In both cases, enhanced navigational accuracy could result from suppression of task-irrelevant information, rather than the enhancement of task-relevant information.

Our data supports the presence of strong local inhibition in CA1 at specific phases of theta (*Figures 2F and 3D*). The duration of this inhibition (~25 ms) is equivalent to approximately 1/5 of a 125 ms (8 Hz) theta cycle, long enough to impact encoding or retrieval functions, but precise enough to avoid disrupting the entire cycle. This suggests a simple, CA1-specific mechanism could be sufficient to explain our behavioral results. According to our proposed mechanism, suppression of inputs carrying information about the current state of the world improves performance in the retrieval

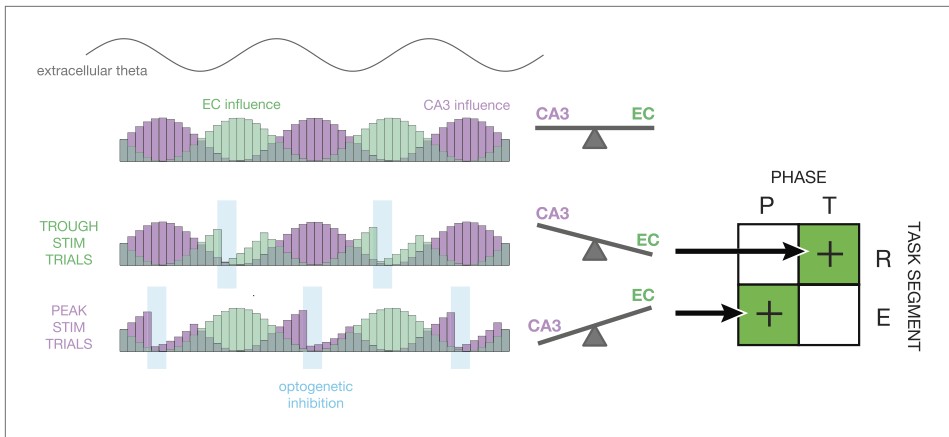

**Figure 6**. Proposed mechanism. This diagram illustrates how the relative influence of CA3 vs entorhinal cortex (EC) inputs to CA1 could explain the experimental results. At the top, a sine wave indicates the phase of theta. Below, the purple and green histograms show the fluctuating influence of CA3 and EC on CA1 on each cycle. Levels represent averages; on individual cycles, one or the other may dominate (***Colgin et al., 2009***). When optogenetic inhibition is triggered on the trough of theta (T), it tends to reduce firing rates in CA1 during periods of high EC influence. This tips the balance in favor of CA3, thereby improving performance during periods of retrieval (R). When optogenetic inhibition is triggered on the peak of theta (P), it tends to reduce firing rates in CA1 during period of high CA3 influence. This tips the balance in favor of EC, thereby improving performance during periods of encoding (E). On the whole, our closed-loop manipulation may improve performance by reducing the influence task-irrelevant inputs as a function of both theta phase and maze region.

segment, whereas suppression of inputs carrying information about past states improves performance in the encoding segment.

Although a local mechanism can provide the simplest explanation of our results, we cannot rule out a mechanism that involves changes in inter-regional coupling strength without simultaneous recordings from the relevant downstream areas. The navigation task used in this study must engage a wide network of brain regions, including those that receive monosynaptic projections from CA1. Besides projections back to the entorhinal cortex, the major cortical output of dorsal CA1 is retrosplenial cortex (***Cenquizca and Swanson, 2007***). Retrosplenial cortex is known to be important for spatial working memory (***Vann and Aggleton, 2004***; ***Keene and Bucci, 2009***; ***Vann et al., 2009***), and changes in coupling between hippocampus and this region could affect performance on the present task. There is also a projection from CA1 to prefrontal cortex, although this originates primarily from the ventral regions (***Swanson, 1981***; ***Cenquizca and Swanson, 2007***). It is possible that our optogenetic manipulation is affecting prefrontal-dependent decision-making via ventral CA1 (***Colgin, 2011***; ***Schmidt et al., 2013***; ***O'Neill et al., 2013***), subicular projections (***Jay and Witter, 1991***), or multisynaptic pathways. Our manipulation could also be exerting long-range effects through the actions of projecting interneurons, a small fraction of which are parvalbumin-positive (***Jinno, 2009***).

Coupling strength between CA1 and medial prefrontal cortex is known to depend on task phase (***Jones and Wilson, 2005***), and it is therefore possible that phase-specific inhibition of CA1 could have a differential impact on CA1–mPFC synchrony during the encoding and retrieval segments. However, if a change in inter-regional coupling strength is at play, we might expect the impact of our optogenetic manipulation to be most pronounced during the retrieval segment, when cortical regions involved in behavioral guidance are likely to access the hippocampal representation of space. In this case, it is not clear that phase-specific stimulation of the hippocampus in the encoding segment should also affect performance. Although our observation of a double-dissociation suggests that the main mechanism is local to the hippocampus, disruption of CA1–mPFC coupling during the encoding segment could minimize cross-trial interference, also leading to enhanced performance.

The influence of alternative behavioral strategies must also be considered. Mice could employ multiple strategies for completing the task, such as an egocentric, hippocampus-independent strategy

based on turn direction or an allocentric, hippocampal-dependent strategy based on an internal map (*Packard et al., 1989*; *Floresco et al., 1997*). It is possible that inhibiting the hippocampus biases the mouse toward an egocentric strategy, which happens to improve performance. If this were the case, again, we would not expect to see such a striking double-dissociation effect in our results. We would instead predict that the same phase of stimulation would enhance performance in both the encoding and retrieval segments. Further evidence against a 'strategy switching' mechanism could come from an experiment in which inhibition was activated at the optimal phase for *both* the encoding and retrieval segments on individual trials. If stimulation is merely invoking a change in strategy, we would not expect to see additive effects; that is, stimulation in the encoding phase would be sufficient to reach peak performance. However, if combined encoding and retrieval stimulation improved performance more than one or the other in isolation, it would suggest that specific effects on encoding and retrieval operations are at play.

The analysis of types of errors in *Figure 5E* makes it clear that encoding and retrieval are continuous processes. They are not necessarily confined to the 'encoding' and 'retrieval' segments specific to this task. The mice are actually performing at least two tasks concurrently, one which involves entry into the cued reward arm (the trained task) and one which involves acting based on the outcome of the previous trial (an untrained task). Could optogenetic stimulation merely serve to 'reset' the system, reducing interference between trials? We consider this possibility unlikely, due to the fact that the effects of stimulation at different phases depended on the track segment in which it occurred. This double-dissociation indicates that stimulation at the appropriate phase allowed mice to more accurately update and retrieve knowledge of the upcoming reward location, rather than simply suppress the influence of the previous trial. In addition, optogenetic stimulation did not impair performance for 'easy' trials, in which the cued reward location was consistent with animals' tendency toward a 'win–stay, lose–switch' strategy (*Figure 5F*). If stimulation brought mice back to a naïve 'baseline' state, we would expect them to make more errors when inhibition is recruited on these trials. Overall, this analysis highlights the fact that encoding and retrieval cannot be considered discrete states that depend on the task at hand, and are instead occurring continuously as animals explore their environment.

Our results indicate that optogenetic inhibition of CA1 serves to *improve* performance across all mice tested. Our initial hypothesis was that the effects of stimulation would depend on both the task segment and phase, but we were unsure if they would be beneficial or punitive. Given that we are recruiting inhibition, and thereby suppressing CA1 output, one might expect the behavioral impact on a hippocampal-dependent task to be negative. Recruiting inhibition during the 'retrieval' phase should impair performance in the retrieval segment, whereas recruiting inhibition during the 'encoding' phase should impair performance during the encoding segment. The fact that we instead observed only enhanced performance (rather than enhancement for one phase of stimulation and impairment for the other) may be explained by a floor effect. Baseline performance was modest at the start of testing (*Figure 1A*, *Table 1*), which makes it unlikely that we could observe a significant impairment, even if it did exist. Mice were strongly influenced by the outcome of the previous trial (*Figure 5E*), which explains why their accuracy on the trained task is only slightly (but significantly) above chance. Our phase-specific optogenetic intervention helps them overcome this bias, especially in the case of trials in which they are required to switch arms after receiving reward (*Figure 5F*). However, even for trials in which the reward location was consistent with animals' intrinsic biases, stimulation did not interfere with performance. It is possible that higher light intensities, alternate fiber placements, or a different target phase could have created the conditions necessary to negatively impact behavior.

Revealing a convincing mechanistic explanation for the behavioral effects seen in this study will require further investigation. The present results justify more extensive inquiry along these lines by providing evidence that processes related to encoding new information and retrieving stored information are most active at different phases of theta. This hypothesis was originally based on a computational model (*Hasselmo et al., 2002*), which was later supported by correlative evidence (*Manns et al., 2007*; *Colgin et al., 2009*; *Douchamps et al., 2013*; *Newman et al., 2013*). We advance this line of investigation through the use of a closed-loop optogenetic intervention that allowed us to interact with the hippocampus at specific phases of theta on a trial-by-trial basis. As the tools for closed-loop control become more accessible, experiments that couple precise stimulation to internal state variables have the potential to enhance our understanding of a wide range of topics related to the study of neural systems.

# Materials and methods

## Animals

All mice were male parvalbumin-Cre (PV-Cre) heterozygotes, derived from PV-Cre BAC transgenics back-crossed into a C57BL/6J line (Jackson Laboratory strain B6; 129P2-Pvalbtm1(cre)Arbr). Mice were 8–12 weeks old at the start of training (mean age = 10.8 ± 1.5 weeks) and 10–15 weeks old at the time of surgery (mean age = 13.5 ± 2.1 weeks). Animals were individually housed and maintained on a 12-hr light/dark cycle (lights on at 7:00 AM). All experimental procedures and animal care protocols were approved by the Massachusetts Institute of Technology Institutional Animal Care and Use Committees and were in accordance with NIH guidelines.

## Task structure

The task was adapted from that used in a previous study (*Jones and Wilson, 2005*). The track consisted of two T-mazes placed end-to-end to form an 'H' shape, with movable gates at both choice points. When running toward the sample arms, the location of the gate forced the mouse in one direction or the other. No reward was delivered in the sample arm, but mice were required to reach the end of it in order to initiate a new trial. When running in the opposite direction, mice could choose between one of two 'free choice' arms. Reward was only delivered if the mouse entered the free choice arm closest to the most recent sample arm it had visited. Rewards consisted of one 14 mg sugar pellet (Bio-Serv, Flemington, NJ; product #F05684), and were always preceded by a 2 kHz tone lasting 250 ms, triggered by entry into the reward zone.

The track was made from laser-cut acrylic, with transparent walls and a black floor. Distal cues were provided by three large black curtains with high-contrast patterns in the center and the experimenter's body, which remained in a consistent location across days. IR sensors were used to monitor entry and exit from different regions of the track. An Arduino sent information about IR beam breaks to a computer running custom software written in Processing (https://github.com/jsiegle/t-maze). The experimenter manually moved the 'forced choice' gates at the start of each trial according to a sequence generated randomly by the behavior computer. If mice were biased toward one reward arm, the probability of reward appearing in the opposite arm increased according to the following equation: $P$(reward in left arm) = $P$(mouse chose right arm during last 12 trials).

## Behavioral training

Prior to the start of training, mice were restricted to 2–3 g of dry food per day, with unlimited access to water. Training began with 4–6 days of habituation, during which mice freely explored the track while receiving reward in both the choice and sample arms. Next, a period of 'forced choice' training began, in which a gate always forced the mice in the correct direction at each choice point. After 5–6 days of forced choice training, a 'free choice' condition was added, in which mice were allowed to make incorrect decisions. Subsequent sessions typically consisted of 10–15 min of forced choice training, followed by 15–20 min of free choice training. Mice received free choice training for 0–10 days before surgery, and 14–26 days prior to the start of behavioral testing.

## AAV vectors

We used AAV-5 viral vectors containing double-floxed, inverted, open-reading-frame ChR2 (H134R variant) coupled to EYFP and driven by the EF1α promoter.

## Fiber optic–electrode implants

Implants were constructed according to the procedure described in *Voigts et al. (2013)*. Design files can be found on GitHub (https://github.com/open-ephys/flexdrive), and assembly instructions are hosted on the Open Ephys wiki (https://open-ephys.atlassian.net/wiki/display/OEW/flexDrive). The base of the drive consisted of two stainless steel cannulae with their centers 3.6 mm apart. Each cannula held four electrodes spaced in a ring around a central fiber optic cable (240 micron core diameter, 0.51 NA, Edmund Optics, Barrington, NJ; part #02-531). The fiber optic cables protruded 0.9 mm past the end of each cannula. Electrodes were made from 12.5 µm polyimide-coated nichrome wire (Kanthal, Hallstahammar, Sweden), twisted and heated to form tetrodes (*Nguyen et al., 2009*). Individual electrodes were gold plated to an impedance of 200–400 kΩ.

## Surgical procedure

Mice were anesthetized with isofluorane gas anesthesia (0.75–1.25% in 1 l/min oxygen) and secured in a stereotaxic apparatus. The scalp was shaved, wiped with hair removal cream, and cleaned with iodine solution and alcohol. Following IP injection of Buprenex (0.1 mg/kg, as an analgesic), the skull was exposed with an incision along the midline. After the skull was cleaned, six steel watch screws were implanted in the skull, one of which served as ground.

Next, a ~1.5 mm-diameter craniotomy was drilled over left hippocampus (2.0 mm posterior to bregma and 1.8 mm lateral to the midline) and the dura was removed. Virus was delivered through a glass micropipette attached to a Quintessential Stereotaxic Injector (Stoelting, Wood Dale, IL). The glass micropipette was lowered through the center of the craniotomy to a depth of 1.2 mm below the cortical surface. A bolus of 1 µl of virus (see details above) was injected at a rate of 0.05 µl/min. After the injection, the pipette was held in place for 10 min at the injection depth before being fully retracted from the brain. The same procedure was then repeated for the opposite hemisphere.

The fiber optic–electrode implant (see details above) was aligned with the two craniotomies, and lowered until the cannulae were flush with the cortical surface. This placed the two fiber optic cables just above the CA1 region of hippocampus (depth of ~0.9 mm). Once the implant was stable, a small ring of black dental acrylic was placed around its base. A drop of surgical lubricant (Surgilube, Fougera Pharmaceuticals, Melville, NY) prevented dental acrylic from contacting the cortical surface. Adhesive luting cement (C&B Metabond, Parkell, Edgewood, NY) was used to further affix the implant to the skull. Once the cement was dry, the scalp incision was closed with VetBond (3M, Saint Paul, MN), and mice were removed from isoflurane.

Following 2–4 days of recovery, electrodes were lowered to their final location over the course of 2–3 weeks. Once stimulation began, electrodes were not adjusted.

## Electrophysiology

On testing days, the track was wiped with an anti-static liquid (Staticide, ACL, Chicago, IL) and cleared of all debris. Electrophysiological data was recorded with the Open Ephys platform (http://open-ephys.org), an open-source data acquisition system based on Intan amplifier chips (http://www.intan-tech.com). Tetrode signals were referenced to ground, filtered between 1 and 7500 Hz, multiplexed, and digitized at 30 kHz on the headstage (design files available at https://github.com/open-ephys/headstage/tree/master/1x32_Omnetics_Standard). Digital signals were transmitted over a 12-wire cable counter-balanced with a system of pulleys and weights. Mouse location was determined via IR gates at behaviorally relevant points along the track and an overhead camera monitoring a red LED mounted on the headstage.

## Stimulation protocol

Online feedback was delivered using the Open Ephys GUI (full source code available at https://github.com/open-ephys/GUI). The trigger channel was filtered between 4 and 12 Hz (2nd-order Butterworth) and sent to a 'Phase Detector' module. When the mouse entered the stimulation segment (either one of two sample arms for 'encoding' sessions or the central arm on forward and reverse trajectories for 'retrieval' sessions), the Phase Detector emitted trigger events when the signal reached a local maximum ('peak') or local minimum ('trough'). Trials of each type ('peak', 'trough', or 'blank') were randomly interleaved with equal probability. Stimulation was triggered via a USB connection to a Pulse Pal (https://sites.google.com/site/pulsepalwiki/), and consisted of 10 ms light pulses from a Plexon PlexBright LED (465 nm, ~50 mW/mm$^2$).

## Histology

At the end of training, electrodes sites were lesioned with 15 µA of current for 10 s. Mice were transcardially perfused with 100 mM PBS followed by 4% formaldehyde in PBS. Brains were post-fixed for at least 18 hr at 4°C. 60 µm sections were mounted on glass slides with Vectashield (Vector Laboratories, Burlingame, CA), coverslipped, and imaged with an upright fluorescent microscope. Viral expression was confirmed by observing EYFP expression beneath the fiber optic lesions in CA1 of all animals. Expression spread ~2 mm along the length of the dorsal hippocampus, primarily in CA1, but also in the lateral portion of CA3. Labeling was strongest in the hippocampal cell layer, where parvalbumin-positive cells have the densest projections. There was also expression in overlying cortex. No

expression was observed in the dentate gyrus. In all animals, the lesion corresponding to the electrode used to trigger stimulation was identified.

## Data analysis

All data analysis was performed using custom Matlab scripts (https://github.com/open-ephys/analysis-tools). Spike activity was extracted offline by thresholding the 300–6000 Hz bandpassed signal. Units were clustered with Simple Clust software (https://github.com/moorelab/simpleclust), based on peak heights and regression coefficients for individual waveforms. Spikes were aligned to light pulses using event timestamps, or to the phase of LFP theta.

To determine the actual phase of theta without the phase shift associated with online filtering, we filtered the wideband, full-sample-rate data offline using the Matlab 'filtfilt' function (2nd-order Butterworth, 4–12 Hz bandpass). We used the angle of the Hilbert-transformed signal to compute the phase in degrees ($-180°$–$180°$, peak at $0°$). Spectral analysis was performing using the Chronux toolbox (http://www.chronux.org), using multitaper methods (time–bandwidth product = 2, number of tapers = 3).

Behavioral analysis was limited to the first 150 trials performed in each condition (encoding stimulation vs retrieval stimulation). Trials were grouped by stimulation type (blank, peak-triggered, or trough-triggered) and the responses (0 = correct choice, 1 = incorrect choice) were averaged. p-values for individual mice were computed using the probability density function of the binomial distribution, with $N$ = the number of trials of a given type and $p$ = baseline accuracy. The probability that a double-dissociation would occur by chance was computing using a bootstrap method with randomized trial labels (10,000 iterations).

## Acknowledgements

We thank K Kim and R LeCoultre for their help running experiments. We are grateful for G Hale, J Sanders, J Voigts, D Meletis, M Carlén, and R Harrison for technical assistance. We thank members of the Wilson Lab for their comments on the figures and manuscript. This study was supported by a grant from the NIH to M Wilson and NDSEG and NRSA fellowships to J Siegle.

## Additional information

### Funding

| Funder | Grant reference number | Author |
| --- | --- | --- |
| National Institutes of Health | National Research Service Award, 1F31MH098508 | Joshua H Siegle |
| NDSEG Fellowship Program | | Joshua H Siegle |
| National Institutes of Health | 1-RO1-GM104948-02 | Matthew A Wilson |

The funders had no role in study design, data collection and interpretation, or the decision to submit the work for publication.

### Author contributions

JHS, Conception and design, Acquisition of data, Analysis and interpretation of data, Drafting or revising the article; MAW, Conception and design, Analysis and interpretation of data, Drafting or revising the article

### Ethics

Animal experimentation: This study was performed in strict accordance with the recommendations in the Guide for the Care and Use of Laboratory Animals of the National Institutes of Health. All of the animals were handled according to approved institutional animal care and use committee (IACUC) protocols of MIT. The protocol was approved by the MIT Committee on Animal Care (#0511-044-14). All surgery was performed under isoflurane anesthesia, and every effort was made to minimize suffering.

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
