## [Decision Letter]

Thank you for sending your work entitled “Enhancement of encoding and retrieval
functions through theta-phase-specific manipulation of hippocampus” for
consideration at *eLife*. Your article has been favorably evaluated by
Eve Marder (Senior editor) and 3 reviewers, one of whom is a member of our Board of
Reviewing Editors.

The Reviewing editor and the other reviewers discussed their comments before we reached
this decision, and the Reviewing editor has assembled the following comments to help you
prepare a revised submission.

This is a fascinating and important article showing a striking double dissociation of
behavioral effects caused by optogenetic activation of parvalbumin-positive interneurons
in hippocampal region CA1. The activation of PV neurons causes highly temporally
specific spiking in the PV cells and a period of about 25 msec of inhibition of spiking
in pyramidal cells. The behavior was tested in an end to end T maze where rats had to
match the start location A vs. B (described as the encoding segment) with a response to
A or B arm after running along a shared stem (described as the retrieval segment).
Effects of different stimulation on the peak or trough of theta were evaluated when a
rat was in the encoding or retrieval portion of the task. Behavioral performance was
enhanced when peak-triggered stimulation was delivered in the encoding segment (falling
on the falling slope of theta and presumably reducing CA3 influence during encoding,
thereby reducing interference from retrieval during encoding). Behavioral performance
was also improved when stimulation was trough triggered in the retrieval segment
(falling on the rising slope of theta and presumably reducing EC influence, thereby
reducing distraction from external input during retrieval). This double dissociation is
very striking and theoretically significant and highly deserving of publication in a
high impact journal. The experiment is very technically sophisticated and has addressed
important questions of behavioral controls (determining whether behavior changed during
stimulation) and questions about neurophysiological changes during stimulation
(primarily there was a strong reduction in pyramidal cell spiking activity during the
optogenetic activity). The specificity of their effect on behavior depends not on the
degree of activity in CA1, but the content of the activity, as they note in the
Discussion.

Major concerns:

1) The main issue is that there is some confusion about the phase of theta at which the
authors delivered the light stimulation. First, there is a bit of a disconnect between
the author's review of the literature, which emphasizes differences between the
peak and the trough of theta oscillations as recorded in the hippocampal fissure, and
the author's stimulation protocol, which targeted stimulation at either the falling
or rising phase of theta at sites other than the hippocampal fissure. The authors should
more explicitly link their protocol to the literature and provide the rationale for why
their stimulation protocol differed from what they led the reader to expect. Second, the
anatomical position of the electrode that the authors used to record theta differed
across the four mice (one each in: stratum radiatum at the CA1/CA3 border, stratum
pyramidale in CA1, unspecified layer of dentate gyrus, and unspecified area and layer of
cortex above CA1). As a result, the phase of theta at which the authors delivered light
stimulation would differ across mice. For example, the “peak” of theta
would be similar between the cortex above CA1 and CA1 pyramidal layer, but these peaks
would not align with the peaks of theta from the electrode in stratum radiatum. None of
these peaks would align with the peaks of theta recorded in dentate gyrus. The authors
discuss this issue to some extent in the discussion, but they give the impression that
only one mouse would differ from the other three. Instead, across the four mice, the
authors effectively used three different definitions of theta phase. Indeed, given this
variability, the relative consistency of the effects of stimulation on behavior are
puzzling. Meaningful interpretations of the results will depend on the extent to which
the authors can resolve this puzzle.

2) A potentially related concern is that the straightforward predictions should have
been that inactivation of principal cell activity at the theta trough (the putative
“encoding phase” of theta) during the encoding phase would impair memory,
whereas inactivation at the peak (the putative “retrieval phase” of theta)
during retrieval would impair memory. These were not observed. Instead, they observed
somewhat of the converse. We think they should acknowledge, rather than avoid, the
straightforward predictions.

3) Some discussion of the marginal performance accuracy of the mice should be included.
After substantial training, they perform barely above 60% correct, hardly compelling
evidence that mice really learn this task. On the other hand, this is a good baseline
for improvement by stimulation, as observed. Also, the authors should consider that the
poor performance is likely because of the high degree of interference between the many
repeated left- and right-turn trials, consistent with the favored interpretation that
stimulation at the right time may reduce interference of competing memories, which could
be viewed as the principal variable in controlling performance.

4) How can they exclude stimulation produced alterations in high frequency oscillations
as a confound?

---

## [Author Response]

We appreciate the reviewers’ enthusiasm for our study, and thank them for their
helpful feedback. The changes made in response to their comments have substantially
improved the quality of our manuscript. Most importantly, we have updated our analysis
of stimulation times to use absolute, rather than relative, phase. A new data panel,
which shows consistent phase of stimulation across animals, makes the main result of our
experiments much more interpretable. We have also added additional figure panels that
document the types of errors mice make in our task. This helps to clarify the reasons
our optogenetic stimulation benefitted behavior. Below, we describe how we have
addressed the specific concerns put forward by the reviewers.

*1) The main issue is that there is some confusion about the phase of theta at
which the authors delivered the light stimulation. First, there is a bit of a
disconnect between the author's review of the literature, which emphasizes
differences between the peak and the trough of theta oscillations as recorded in the
hippocampal fissure, and the author's stimulation protocol, which targeted
stimulation at either the falling or rising phase of theta at sites other than the
hippocampal fissure. The authors should more explicitly link their protocol to the
literature and provide the rationale for why their stimulation protocol differed from
what they led the reader to expect*.

We have added two sentences to the Introduction that address this point:

“Our stimulation occurred relative to the phase of locally recorded theta on the
trigger electrodes, rather than the phase at the hippocampal fissure, to which much of
the previous literature uses as a landmark (3; 21;
15). However, post-hoc
analysis revealed that light pulses were delivered at similar absolute phases across
animals.”

*Second, the anatomical position of the electrode that the authors used to record
theta differed across the four mice (one each in: stratum radiatum at the CA1/CA3
border, stratum pyramidale in CA1, unspecified layer of dentate gyrus, and
unspecified area and layer of cortex above CA1). As a result, the phase of theta at
which the authors delivered light stimulation would differ across mice. For example,
the “peak” of theta would be similar between the cortex above CA1 and
CA1 pyramidal layer, but these peaks would not align with the peaks of theta from the
electrode in stratum radiatum. None of these peaks would align with the peaks of
theta recorded in dentate gyrus. The authors discuss this issue to some extent in the
discussion, but they give the impression that only one mouse would differ from the
other three. Instead, across the four mice, the authors effectively used three
different definitions of theta phase. Indeed, given this variability, the relative
consistency of the effects of stimulation on behavior are puzzling. Meaningful
interpretations of the results will depend on the extent to which the authors can
resolve this puzzle*.

The reviewers are correct to point out that meaningful interpretation of our behavioral
results depends on our ability to measure stimulation phase relative to some absolute
landmark within the theta cycle. In response, we have completed a more comprehensive
analysis of both the histological and electrophysiological data that shed light on this
issue. These efforts have produced a new data panel (Figure 4), as well as additional text in the Results section.

The general conclusion is that, in all instances we can measure, the phase of peak- and
trough-triggered stimulation relative to a ’landmark’ event; the peak of
high gamma power is remarkably consistent across mice. In addition to showing the
relationship between stimulation times and high gamma power for these two conditions, we
also show images of the precise locations of each electrode lesion. Because absolute
theta phase was not captured by our previous descriptions and analysis, we think readers
will find the current presentation much more informative. In addition, the similarity of
stimulation times relative to the high gamma peak (except in one animal, in which we
were unable to measure this variable) supports the hypothesis that our consistent
behavioral results are due to the precision of our phase-specific intervention.

*2) A potentially related concern is that the straightforward predictions should
have been that inactivation of principal cell activity at the theta trough (the
putative “encoding phase” of theta) during the encoding phase would
impair memory, whereas inactivation at the peak (the putative “retrieval
phase” of theta) during retrieval would impair memory. These were not
observed. Instead, they observed somewhat of the converse. We think they should
acknowledge, rather than avoid, the straightforward predictions*.

The Discussion section now directly addresses the lack of any apparent decrease in
behavioral performance as a result of inactivation:

“Our initial hypothesis was that the effects of stimulation would depend on both
the task segment and phase, but we were unsure if they would be beneficial or punitive.
Given that we are recruiting inhibition, and thereby suppressing CA1 output, one might
expect the behavioral impact on a hippocampal-dependent task to be negative. Recruiting
inhibition during the ‘retrieval’ phase should impair performance in the
retrieval segment, whereas recruiting inhibition during the ‘encoding’
phase should impair performance during the encoding segment.”

We have also expanded our treatment of the supposed ‘floor effect’ that
may explain why we observed behavioral enhancement at certain phases, without a
corresponding impairment at the opposite phases:

“Mice were strongly influenced by the outcome of the previous trial (Figure 5), which explains why their accuracy on the
trained task is only slightly (but significantly) above chance. Our phase-specific
optogenetic intervention helps them overcome this bias, especially in the case of trials
in which they are required to switch arms after receiving reward (Figure 5). However, even for trials in which the reward location
was consistent with animals’ intrinsic biases, stimulation did not interfere with
performance. It is possible that higher light intensities, alternate fiber placements,
or a different target phase could have created the conditions necessary to negatively
impact behavior.”

*3) Some discussion of the marginal performance accuracy of the mice should be
included. After substantial training, they perform barely above 60% correct, hardly
compelling evidence that mice really learn this task. On the other hand, this is a
good baseline for improvement by stimulation, as observed. Also, the authors should
consider that the poor performance is likely because of the high degree of
interference between the many repeated left- and right-turn trials, consistent with
the favored interpretation that stimulation at the right time may reduce interference
of competing memories, which could be viewed as the principal variable in controlling
performance*.

We have conducted an in-depth analysis of the types of errors to which mice are prone,
and have found that, indeed, there is a high degree of interference between adjacent
trials. Mice display a striking—but unsurprising—bias toward risk
aversion. If they receive reward on one trial, they are more likely to visit the same
reward arm on the next trial, regardless of the spatial cue. If they make a mistake, and
no reward is given, they are more likely to choose the alternate arm on the next trial.
This means the mice are essentially performing at least two tasks
concurrently—one that involves the cued reward location, and one that involves
the outcome of the previous trial. These findings (which stem from the analysis of
baseline trials only) are summarized in a new data panel, Figure 5.

When we add our optogenetic manipulation, we see that the main effect of stimulation at
the optimal phase is to allow mice to overcome their bias toward returning to the
rewarded arm (Figure 5). We do not see a
corresponding decrease in performance on trials in which mice correctly choose the
previously rewarded arm, indicating that the light pulses are not simply increasing
response variability.

We have added new paragraphs to the Results section and Discussion section describing
and analyzing these findings. Additional interpretations of our results in light of
these findings have been added to the Discussion.

4) How can they exclude stimulation produced alterations in high frequency
oscillations as a confound?

It appears that the changes in low-gamma-range oscillations are primarily due to leakage
from the beta band (16-25 Hz), which is itself explained by the shape of the evoked
response to the light pulse. The duration of this response (∼50 ms) will cause a
peak to appear in the beta/low gamma range following spectral analysis. As far as we can
tell, this does not imply that stimulation induces a change in the resonant properties
of the local circuit. In addition, because we see similar changes in the power spectra
for the encoding and retrieval, it does not appear that the oscillations we recorded can
explain our behavioral result. There was a double-dissociation for the impact of our
manipulation on behavior, but not on locally recorded high-frequency oscillations.

We have added the following sentences to the results section, describing this new
interpretation of our spectral analysis:

“There was also an increase in power in the low-gamma band (25-35 Hz) for both
peak and trough stimulation, but this was associated with a much stronger peak in the
beta band (16-25 Hz), which may have affected the low-gamma band via spectral leakage.
Based on the shape of the evoked response to each optogenetic stimulus, it appears that
these effects are due to the frequency content of the average waveform, rather than
non-phase-aligned induced power in different frequency bands (Figure 4). Aligning the local field potential to the start of each
light pulse revealed a large deflection, 200-400 µV in amplitude. The shape of the
average response accounts for both the shifts in theta frequency (based on the location
of the subsequent peak), and the beta-range power increases (due to ∼50 ms
deflections).”